

# Impact of the the COVID-19 pandemic on a queen conch (*Aliger gigas*) fishery in The Bahamas

Nicholas D. Higgs

Cape Eleuthera Institute, Rock Sound, Eleuthera, Bahamas

## ABSTRACT

The onset of the coronavirus (COVID-19) pandemic in early 2020 led to a dramatic rise in unemployment and fears about food-security throughout the Caribbean region. Subsistence fisheries were one of the few activities permitted during emergency lockdown in The Bahamas, leading many to turn to the sea for food. Detailed monitoring of a small-scale subsistence fishery for queen conch was undertaken during the implementation of coronavirus emergency control measures over a period of twelve weeks. Weekly landings data showed a surge in fishing during the first three weeks where landings were 3.4 times higher than subsequent weeks. Overall 90% of the catch was below the minimum legal-size threshold and individual yield declined by 22% during the lockdown period. This study highlights the role of small-scale fisheries as a 'natural insurance' against socio-economic shocks and a source of resilience for small island communities at times of crisis. It also underscores the risks to food security and long-term sustainability of fishery stocks posed by overexploitation of natural resources.

## INTRODUCTION

Subsistence fishing has played an integral role in sustaining island communities for thousands of years, especially small islands with limited terrestrial resources (*Keegan et al., 2008*). In subsistence fisheries the catch goes directly to providing food for the family or wider community of the fishers, rather than being sold to merchants for distribution to wider markets as in artisanal fisheries (although see *Schumann & Macinko, 2007*) for a more nuanced discussion of this definition). Despite this basic contribution to food security, subsistence fisheries are frequently overlooked in official fisheries statistics (*Pauly, 2018*) and economic analyses (*Bevilacqua et al., 2019*). This is partly because subsistence fisheries effort tends to be widely dispersed and hard to monitor, but also because it is often a secondary occupation that is not captured in official census data and may be temporally inconsistent.

The Bahamas provides an interesting case-study for examining the role that subsistence fishing plays in the food security of coastal communities. It ranks second in the Caribbean region for dependency on food imports (*Shik et al., 2018*), so is highly susceptible to

Corresponding author
Nicholas D. Higgs,
nickhiggs@ceibahamas.org

potential disruptions to food supply chains. Despite being listed as a high-income country by the World Bank, pronounced economic inequality means that there is an elevated level of food insecurity, with ~40% of people reporting worry about having enough food to eat and 29% of households reporting that they had run out of food at some point in the preceding year (*Rahming, 2019*). Subsistence fishing is an important food source for island communities, providing an estimated 590 tonnes of seafood to the Bahamian population per year, ~16–33 kg per person per year (*Smith & Zeller, 2016*).

The queen conch, *Aliger gigas* (Linnaeus, 1758) = *Strombus gigas*, *sensu Maxwell et al. (2020)*, is the most culturally significant fishery species in The Bahamas, forming a staple part of the country's diet (*Sherman et al., 2018*). These large and highly prized marine molluscs are slow moving inhabitants of shallow waters, making them a relatively accessible source of seafood protein. Over 40% of people in Bahamian rural communities consume conch at least once per month and another 37% report eating conch on a weekly basis (*Bomhauer-Beins, De Guttry & Ratter, 2019*). However; recent trends in overexploitation by artisanal fisheries have raised serious concerns about the long-term sustainability of their populations and dependant fisheries (*Stoner, Davis & Kough, 2019*).

This study examines patterns of queen conch exploitation by subsistence fishers during the global coronavirus pandemic of 2020 in a rural Bahamian setting. Following the enactment of Emergency Orders to control the spread of the coronavirus pandemic, only essential business were allowed to operate, leaving many unemployed or furloughed (Fig. 1). A weekday curfew to shelter at home was put in place with only essential trips to obtain food and medical supplies allowed. Subsistence fishing was, however, one of the few operations allowed to continue during the weekday curfew. In contrast, artisanal fishers were not allowed to sell catch publicly, effectively limiting fishing activity to subsistence fishers. Nevertheless, after the implementation of the curfew an unusually high number of fishers were observed fishing in the study area.

There is no routine monitoring or data collection for small-scale fisheries in The Bahamas (*Kincaid, 2017*), hindering effective management efforts (*Smith & Zeller, 2016*). The aim of this study is thus to document and quantify the changes fishing activity associated with the COVID-19 pandemic and provide an insight into fishery dynamics in the Caribbean region.

## MATERIALS & METHODS

Conch fishery surveys were carried out along a one-mile section of the northern shore of the Cape Eleuthera peninsula (Eleuthera, Bahamas), stretching from the Cape Eleuthera Institute southeast to the entrance of Kemp's Creek (Fig. 2). This section of narrow beach and rocky shore is a well-documented site for subsistence conch fishing, where fishers without access to boats wade across the shallow bank to collect conch by hand (*Clark, Danylchuk & Freeman, 2005*). A main road abuts the shoreline with three commonly used entry points (sites 1, 2a/b and 3 in Fig. 2), where the catch is processed, and shells discarded in piles. The total area accessible by wading (*i.e.*, standing depth) is ~80 hectares.

A visual survey for discarded conch shells was conducted at low tide three times per week and the location of recently deposited conch shell piles (middens) was recorded. At

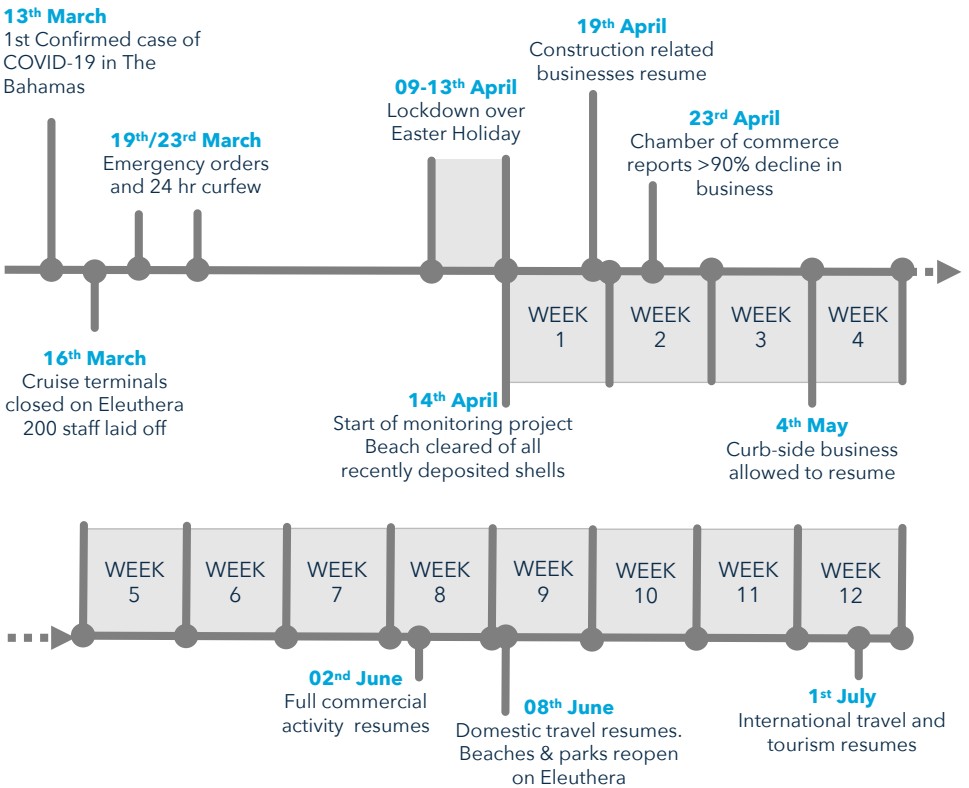

**Figure 1 Timeline of COVID-19 pandemic and associated economic disruption on Eleuthera island, Bahamas in 2020.** Weeks of the lockdown period correspond to weekly collections of conch shells from the subsistence fishery.

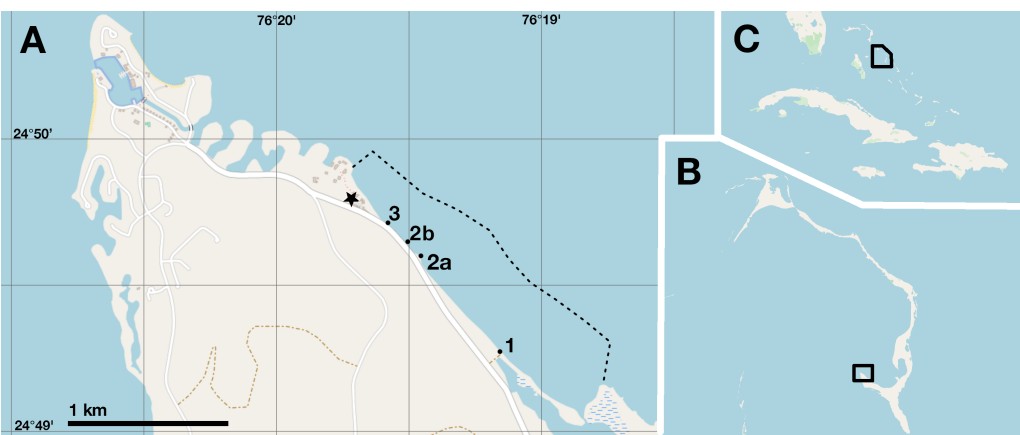

**Figure 2 Study site location.** (A) Cape Eleuthera peninsula showing, numbered collection sites, subsistence fishing grounds demarcated by the dashed line and the Cape Eleuthera Institute (star). (B) Location of study site (black box) on Eleuthera island. (C) Location of Eleuthera island in the Bahamian archipelago (black box). Base map and data from OpenStreetMap and OpenStreetMap Foundation, ©OpenStreetMap contributors.

the end of each week the shells deposited by fishers on or near the shore during that week were collected and brought to the Cape Eleuthera Institute for analysis. Discarded shell middens are assumed to be indicative of fisheries effort. Indeed, fishers do not generally discard shells of dead animals on the fishing grounds in the belief that it drives away live animals (*Thomas & Dodd, 2016*), but instead deposit them in discrete middens. Weekly, haphazard snorkel transects ($2 \times 50$ m) of the fishing grounds revealed occasional discard of isolated shells (<1% of midden numbers) which were from the full shell size spectrum being fished, indicating that this practice was not systematically biasing analyses.

The first collection of shells took place on the morning of the 14th April 2020, following a complete lockdown over the holiday weekend. All shells that still had the periostracum on the outside of the shell and retained pink colouration internally were deemed to have been deposited recently (*Stoner, Davis & Kough, 2019*) and if they showed signs of being killed by fishers were collected. This first cohort represented the accumulated catch over an unknown and undefined period of time leading up to the beginning of the study, hereafter referred to as the "pre-lockdown" period. After the coast was cleared of recently deposited shells in the first collection, all subsequently deposited shells were collected at the end of each week before the weekend curfew, when no fishing activity was permitted and were assumed to represent the week's catch. Weekly shell collections were carried out for 12 weeks following the initial clearing of the coast, hereafter referred to as the "lockdown" period.

Standard morphometric measurements were recorded for all shells that were collected, including total shell length (siphonal length) and shell width as defined by *Martin-Mora, James & Stoner (1995)*. Shells that had a well-defined flared lip (*i.e.*, legal-sized in The Bahamas), aperture length, aperture width and lip thickness were also measured as set out by *Ruga, Meyer & Huntley (2019)*. These measurements were taken using large vernier callipers to the nearest mm, whilst lip thickness was measured using smaller dial callipers to the nearest 0.1 mm as described by *Appeldoorn (1988)*. Total shell length was directly measured for 75% of shells collected. Where shells were damaged or broken, only a subset of these metrics were taken (24% of shells collected). Allometric scaling relationships based on complete shell data were then used to estimate shell length for broken shells, providing reliable shell length data for 99% of conch collected. Only 1% of collected shells ($n = 20$) were too damaged to prevent an estimate of shell length. These damaged shells were only included in analysis of catch abundance.

Shells were classified into size-classes used for this fishery as a proxy for age in the past, allowing for comparison of catch rates with historical data (*Tewfik & Béné, 2000*; *Clark, Danylchuk & Freeman, 2005*). Shell length increases with age for the first ~3.5 years of a conch's life in up to ~200–250 mm length in The Bahamas, after which the shell lip flares out and only increases in thickness with age (*Stoner et al., 2012*). Therefore, conch without a flared lip are separated into three categories of juveniles: (1) small juveniles (SJ) are those less than 150 mm shell length and correspond to the age one and two size classes, while (2) medium juveniles (MJ) are those shells of 150–200 mm length and correspond to the age-3 size class (*Stoner, Davis & Kough, 2019*), and (3) large juveniles (LJ) are those greater than 200 mm length but still do not possess a well-developed lip. The presence of a flared

lip means that the conch is legal-sized in The Bahamas and these are divided into two categories: the "sub-adult" (SA) size class represents those shells with flared lip that is still relatively thin (<4 mm thick), while those with a lip thickness greater than four mm are in the "adult" (AD) category. This terminology is now known to be inaccurate, since size at sexual maturity is highly variable in the queen conch. The *minimum* lip-thickness at which conch reach sexual maturity in The Bahamas is nine mm for males and 12 mm for females, and 15 mm lip thickness is actually a more realistic size at sexual maturity (*Stoner et al., 2012*). Nevertheless, the older size categories are retained in the dataset (Data S1) for comparison with previous studies.

Estimates of cleaned meat obtained from each conch shell were made using empirically derived relationships between shell length and wet meat weight, previously obtained for conch in The Bahamas (*Iversen et al., 1987*), U.S. Virgin Islands (*Wood & Olsen, 1983*) and Puerto Rico (*Appeldoorn, 1988*), which includes changes associated with transition from juvenile to adult growth. The mean value from the three models was used to provide an estimated figure for meat obtained from each conch (Data S1). An indicative figure for the number of individual meals provided by the harvested meat was calculated assuming 198 g conch per person, an average of the mean portion sizes for men and women reported by *Lockhart, Magnusson & Clerveaux (2004)*.

It was not possible to quantitatively nor accurately capture all fishing effort (number of fishers and time spent fishing), however qualitative observations were noted. It was assumed that spatially separate middens (*i.e.*, distinct piles of shells) represent the effort of separate individuals or groups of fishers.

## RESULTS

In total, the shells of 1598 queen conch were collected from the three landing sites during the study: 495 (31%) from the pre-lockdown period and 1103 (69%) collected during the 12 weeks during the lockdown period. Conch catches were relatively high during the first three weeks of the lockdown period, increasing from 204 to 289 conch per week (Fig. 3). Fishing activity then dropped sharply from week 4 onward, coinciding with the partial resumption of commercial activity (Fig. 1). Landings averaged 43 conch per week from weeks 4 through 12, with no shells deposited during weeks 10 and 11 (Fig. 3). Individual middens (*i.e.*, distinct piles of shells) had a mean size of 58 shells, ranging between 14 and 195 conch, throughout the lockdown period (Data S1).

The average shell-length of conch caught during the lockdown period was one cm (6%) shorter (median = 15.8 cm) than those being caught before the lockdown (median = 16.9 cm) (Wilcoxon-Mann–Whitney Test; $W = 351559$, $p = 2.2e-16$). All of the post-lockdown weekly catches had a shorter median shell length compared to pre-lockdown catches, with the exception of the conch taken in week 2 (Fig. 4). Furthermore, there was a significant week-on-week decline in shell length of conch throughout the study period (Jonckheere-Terpstra Test; $J\text{-}T = 395032$, $p = 0.001$).

When the catches are broken down by size-class, the proportion of small juveniles (shell length <15 cm) being captured during the lockdown period more than doubled,
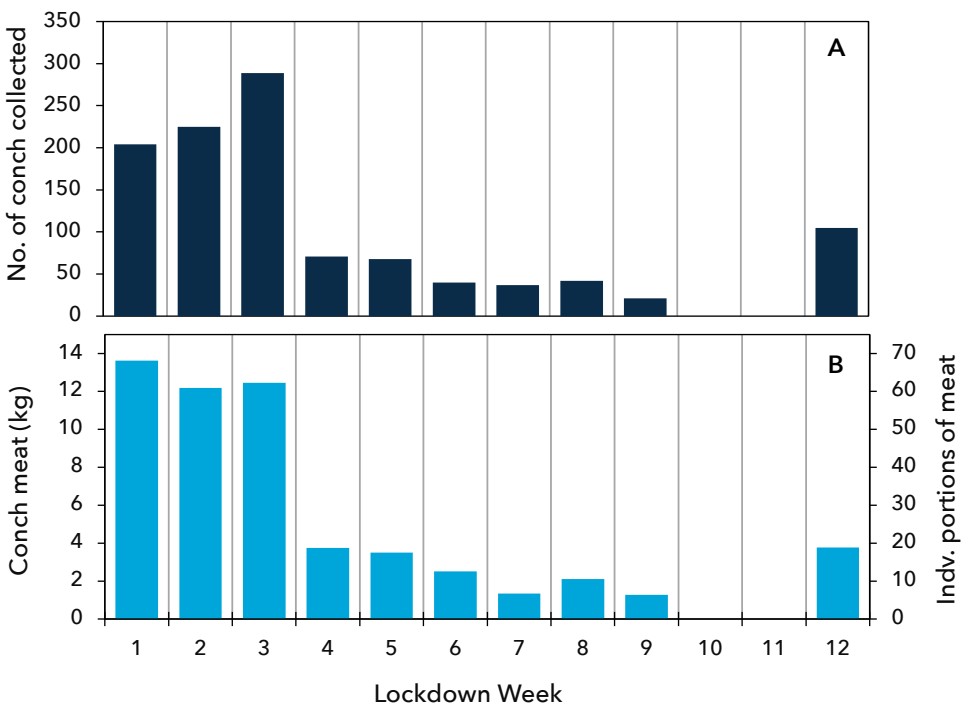

**Figure 3** **Queen conch catch from each week of the lockdown period.** (A) Number of individuals caught. (B) Estimated mass of meat harvested and equivalent individual portions of meat.

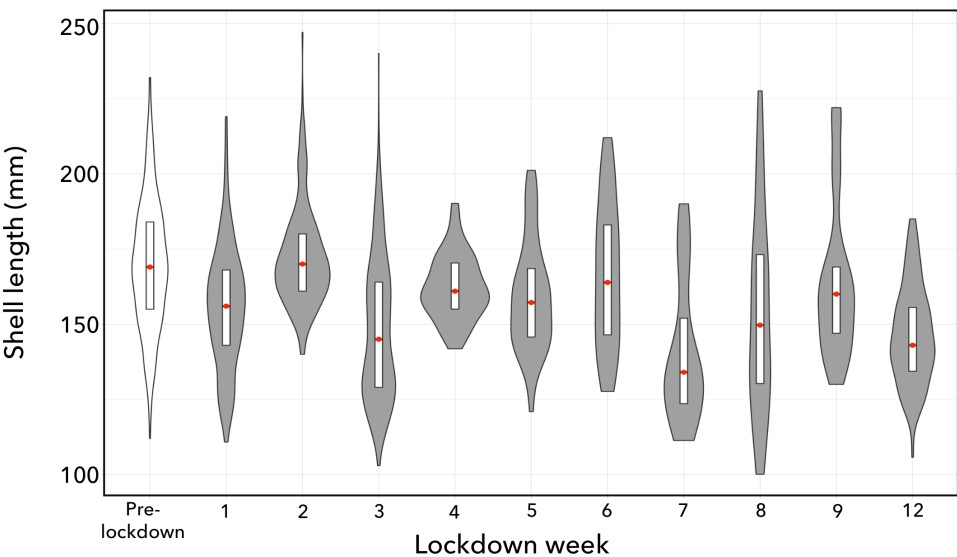

**Figure 4** **Size-frequency distributions of queen conch shells landed by subsistence fishers.** Conch caught in the pre-lockdown period are shown in the unfilled plot and subsequent weeks of the lockdown period shown as filled plots. Frequency curves are normalized to a uniform plot area. Median shell length is indicated by the red points, with white bars showing upper and lower quartile ranges.

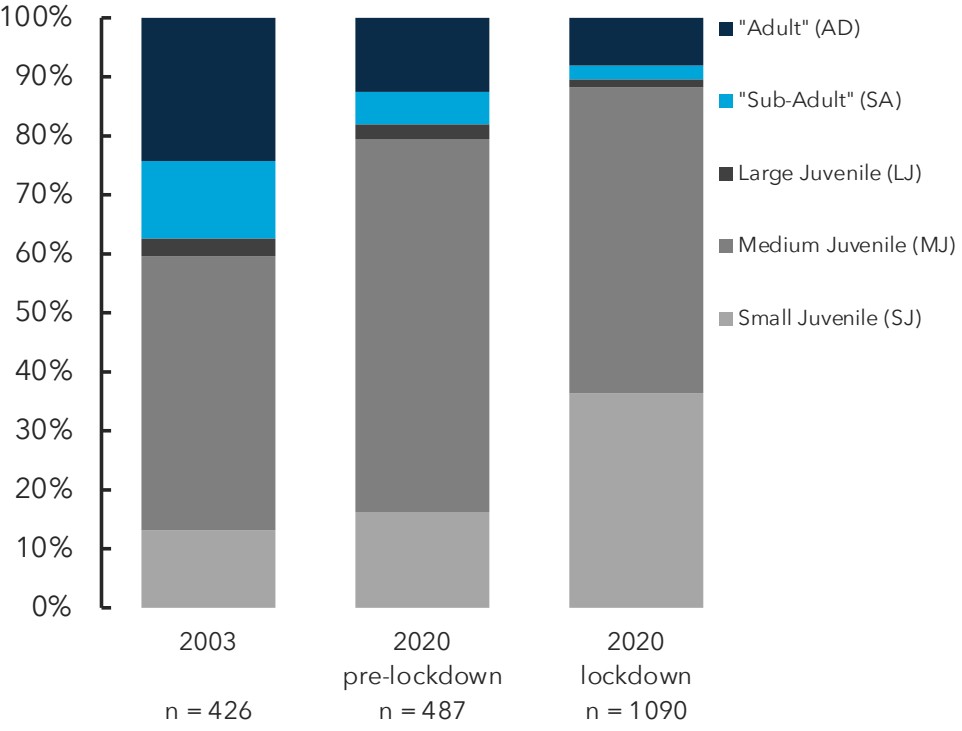

**Figure 5 Queen conch catch broken down by size-class.** Catch proportions are shown for for the pre-lockdown (center) and lockdown periods (right), as well an historic baseline study (*Clark, Danylchuk & Freeman, 2005*), left. Blue bars show legal-sized catch broken down into those with a lip thickness greater than four mm (dark) and less than four mm (light). Grey bars are illegal sized juvenile size classes. See Methods section for more detail of size classifications.

compared to pre-lockdown catches (Fig. 5, lightest-grey bars). Prior to the start of the lockdown period only 18% of the conch being fished had a flared lip (Fig. 5, blue bars), which defines a legally harvestable animal in the Bahamas. The legal-sized proportion of the catch dropped further to only 11% during the lockdown period (Fig. 5). Less than 1% of the total conch landed throughout the study period had a shell lips of ≥15 mm, the recommended shell size for ensuring that animals are sexually mature (*Stoner, Davis & Kough, 2019*).

The decrease in shell length of conch caught during the lockdown period resulted in a 22% reduction in the amount meat obtained from each conch on average, falling from 60 g per conch before the lockdown to 47 g after lockdown. During the first three weeks of the lockdown period the number of conch collected increased by 40%, but the amount of meat obtained from these conch declined by 14% because of the decrease in the average size of conch being taken (Fig. 3).

Fishers were most commonly working in groups of 2-4, never exceeding 10 on the grounds at any one time during the period of the study. Most fishers were adult males, but there were also multi-generational fishing groups and presumed family groups fishing (*i.e.*, men, women and young fishers working together). Fishers used old cooler boxes

that floated to transport collected conch back to shore for processing. Occasionally fishers were observed knocking and discarding conch shells whilst fishing, presumably when their container became full. Snorkel surveys showed no indication that widescale processing was happening offshore, with only isolated shells being found at any one location.

## DISCUSSION

The introduction of coronavirus control measures in a small-island community was associated with a surge in subsistence fishing that lasted for weeks. This period of intense 'panic-fishing' exemplifies the general principle that a shock to food security will induce a resilience response to mitigate short to mid-term consequences, such as the 'panic-buying' and hoarding responses widely observed during the pandemic (*Béné, 2020*). In this instance, subsistence fishing provided an accessible, free source of high-quality traditional food. As soon as economic activity partially resumed there was a steep drop in fishing activity, suggesting a potential causal link that requires further investigation: perhaps a reduction in perceived risk of lacking food or having less available time dedicated to fishing.

The increased level of subsistence fishing during the coronavirus pandemic also exemplifies the role of fisheries in providing a food and income safety-net during periods of extended hardship. During the twelve-week study period a total of 285 portions of conch were harvested from an area of only 80 hectares, enough to provide a meal to every member of the nearby community; 240 persons according to the latest census data. It is unlikely that the catches were distributed evenly, but the catches reported here do not account for the numerous other local sites where conch are fished (*Cash, 2012*), nor for the other fishery species that were contributing to local diets (*Danylchuk, 2005*). The value of fisheries as a natural insurance against socio-economic shocks is also relevant to other severe disruptions such as hurricanes. After the Category 4 hurricane Matthew passed through The Bahamas in 2016, surveys on Andros island showed that fisheries played a key role in community resilience to the damage and disruption caused (*Valdez et al., 2019*).

It is important to consider the degree to which small island communities - and the governments that have a responsibility to them - will be able to rely on fisheries as a safety net in the future, given the long-term economic impacts of the coronavirus pandemic (*Meighoo, 2020*) and increasing intensity of tropical cyclones (*Kossin et al., 2020*) in the region. The results of this study give rise to concern regarding the long-term sustainability of this and other similar fisheries. During the lockdown period there was a sustained decrease in the size of conch being taken and consequently ~90% of the catch fell below the threshold for legal harvest (Fig. 5). Furthermore, both of these metrics (conch size and proportion of legal-sized catch) decreased markedly compared to historical catch data, where there were already concerns about the high degree of fishing juveniles (*Clark, Danylchuk & Freeman, 2005*). This is an example of how certain resilience response strategies may be detrimental to long-term food security, despite providing short-term relief (*Béné, 2020*). This risk is heightened if extraordinary fishing behaviour, *i.e.*, catching very undersized animals during crisis, becomes normalized practice. Such negative feedback loops can have unforeseen ripple effects (*Béné, 2020*); in this case, a delayed reduction in recruitment of juveniles to the wider fishery population.

The particular role of fisheries as a natural insurance against economic and food insecurity has a quantifiable economic benefit that can pay for itself and should be valued accordingly in fisheries management (*Roughgarden & Smith, 1996*). Cape Eleuthera was one of the most productive conch nursery grounds in the Bahamas but has shown declining populations over the last three decades (*Thomas et al., 2015*; *Stoner, Davis & Kough, 2019*). This is consistent with the results here showing that less than 1% of the catch was large enough to ensure sexual maturity (15 mm shell lip thickness). Almost all of the conch taken were immature sub-adults or juveniles that will not have reproduced. The decreasing size of conch fished throughout the study, even as catch volume declined, suggests serial depletion of large individuals in the fishing grounds. Such intense fishing of juveniles as well as all large adults, risks long-term overfishing and stock depletion (*Tewfik et al., 2019*).

It should be noted that the local artisanal fishery operates using small motorboats (as opposed to the 'walk-in' subsistence fishery) and has a much higher proportion of legal-size catch that are generally larger (*Cash, 2012*), owing to their ability to fish a wider area of deeper waters. The taking of small juveniles is directly proportional to the restricted access experienced by many subsistence fishers, limiting them to shallow nursery grounds. This poses acute challenges to mitigating the negative impacts of a socio-economic shock because it is the subsistence fishers who are the most reliant on the fishery in time of crisis, but they are also those who are fishing most unsustainably. The wider fishing grounds including those studied here have been proposed as a marine protected area for fisheries protection for over 15 years (*Danylchuk, 2005*) and have been included for designation in forthcoming legislation. However, there are no specific management plans in place and local fishers have expressed opposition based on perceived threat to fisheries livelihoods (*Thomas et al., 2015*). Appropriate management will have to involve the stakeholders themselves in decision making, in keeping with The Bahamas' implementation of the FAO Small-Scale Fisheries Guidelines (*Kincaid, 2017*).

A full understanding of the social-ecological system (SES) of this subsistence fishery will require extensive consultation with local fishers and the wider community to determine the cultural drivers and causal mechanisms behind the trends reported (*Bomhauer-Beins, De Guttry & Ratter, 2019*). This important work is planned but was not undertaken as part of this study for a number of methodological, practical and ethical reasons. Firstly, this study aimed to document patterns of fisheries exploitation in relation to the coronavirus pandemic and the role of humans as ecosystem agents. Primary importance was placed on obtaining a representative snapshot of fishery changes that would be applicable across small island communities throughout the region, where no monitoring was taking place, but similar dynamics would be at play (*Kincaid, 2017*). Interactions between researchers and fishers may have changed the fishers' behaviour (especially given the level of illegal fishing) that would have given an inaccurate view of the broader reality. On a practical level, the national lockdown meant that there was a legal obligation to cease social interactions, ruling out effective communication with fishers. Most importantly, it was not possible to conduct the necessary social research to the methodological and ethical standards required to protect potential participants (*St John et al., 2014*). From the outset, it was obvious that much of the fishing was illegal in nature and that even anonymised surveys could

put fishers at risk of repercussions (*St John et al., 2016*). For these reasons, anthropology research was delayed until surveys could be undertaken to a rigorous standards ensuring ethical treatment of human participants.

Future work to establish non-crisis fishing practices and levels of fishing effort will be particularly beneficial for establishing a full view of the fisheries response to the coronavirus pandemic. Although it is not possible to accurately establish fishing effort here, the consistently low level of fishing from weeks 4–12 after the implementation of lockdown indicate a likely baseline of effort (Fig. 3). The slight increase of fishing on week 12 may have been compensating for the lack of fishing in the preceding weeks (possibly related to adverse weather) or was related to the resumption of tourism and associated demand for conch. Further evidence that the fishing effort in weeks 1–3 was unusually high can be gained by inference from the shells deposited before the lockdown. During the first three weeks of fishing after lockdown, a high level of fishing effort was about 250 conch per week (Fig. 3). When the shoreline was cleared of all recently deposited shells at the beginning of this study (see methods section), a total of 495 shells were collected, which would only represent 2 weeks catch. However, given the weathered state of some of the shells, this estimate is unlikely. Alternatively, the average of harvesting of 48 conch per week represents a more realistic 10 weeks of catch. Therefore, the high levels of fishing observed at the start of the lockdown period were truly exceptional and coincident with coronavirus mitigation measures.

## CONCLUSIONS

Small-scale subsistence fishing surged in intensity with the introduction of COVID-19 control measures in a rural small-scale fishery, suggesting a reliance on the marine resources as a safety net during times of crisis (*Valdez et al., 2019*). The increase in fishing intensity also corresponded to a higher proportion of illegal-sized catch with smaller animals than pre-lockdown catches. The mean size of individuals landed during lockdown progressively decreased over time, showing a serial depletion of larger animals. These results emphasize the role and value of fisheries as a natural insurance for small island communities. They also highlight potential risks posed by overfishing to the long-term ability of fisheries of providing resilience against future socio-economic shocks. Further work in consultation with local fishers should provide a more complete view of the human dimension of this socio-ecological system (*Aswani, 2019*). Incorporating human dimensions of the system can form the basis of more effective management ensureing long-term sustainability of this culturally valuable resource.

## ACKNOWLEDGEMENTS

I am grateful to Faith Higgs for assistance in the field.

### Funding

Funding and support (staff time and resources) for this work was provided entirely by the Cape Eleuthera Foundation. The funders had no role in study design, data collection and analysis, decision to publish, or preparation of the manuscript.

### Grant Disclosures

The following grant information was disclosed by the author:
Cape Eleuthera Foundation.

### Competing Interests

The authors declare there are no competing interests.

### Author Contributions

- Nicholas D. Higgs conceived and designed the experiments, performed the experiments, analyzed the data, prepared figures and/or tables, authored or reviewed drafts of the paper, and approved the final draft.

### Data Availability

The raw measurements and computed data are available in the Supplementary File.

### Supplemental Information

Supplemental information for this article can be found online at http://dx.doi.org/10.7717/peerj.11924#supplemental-information.

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
