# Peer review of "Impact of the the COVID-19 pandemic on a queen conch (Aliger gigas) fishery in The Bahamas"

_PeerJ, doi:10.7717/peerj.11924_

## Round 0.1 · original submission · Minor Revisions

· Academic Editor

Minor Revisions

We all appreciate your rigorous and important investigation made during the pandemic and the well-written accounts of the study. Please make sure to address all points mentioned by the reviewers for your revision to expedite the publication.

·

Basic reporting

This article seeks to evaluate how an increase in fishing pressure, on a population of queen conch, during the introduction of COVID-19 control measures effects a rural small-scale fishery in The Bahamas.
The article is well written and suitable references are used and good background information is given.
Figures and tables are sufficient and raw data is available.
The results may be reasonably predictable, given the circumstances, but demonstrate how just moderate increases in fishing pressure on a small population of easily fished conch can have a serious impact in a short period of time.

Experimental design

The work seeks to use the situation brought about by the global pandemic and subsequent restrictions imposed in many communities to be able to evaluate the impact upon the conch fishery when subsistence fishing is one of a few activities allowed.
Although this does create a unique opportunity the results could most likely be predicted, although the question remains well define and is potentially meaningful.
Methods are described well and could be repeated.

Validity of the findings

Data is robust and the conclusions are well stated and acceptable

Additional comments

I think it would be helpful are the start to just define what subsistence fishing is, say with respect to artisanal fishing. You touch on this later but some guidance at the start would help in my opinion.
Also consider some wider referencing outside of The Bahamas, for example Tiley et al 2018 suggest that lip thickness should also be used for juvenile conch
J Invertebr Pathol 2018 Jun;155:32-37.
doi: 10.1016/j.jip.2018.04.007. Epub 2018 Apr 24.
Pathology and reproductive health of queen conch (Lobatus gigas) in St. Kitts

Reviewer 2 ·

Basic reporting

1-Basic reporting

-Clear and unambiguous, professional English used throughout. Literature references, sufficient field background/context provided.

Professional English used through, let references and background could be improved.

-Professional article structure, figures, tables. Raw data shared.
Self-contained with relevant results to hypotheses.

Reporting is clear and well written and supported to the extent possible in the underreported topic of subsistence fishing. The structure of the article is professional and raw data is shared.

Experimental design

2-Experimental Design

-Original primary research within Aims and Scope of the journal.
Subsistence fishing data is rare in the Bahamas so this is unique snapshot of such data collected opportunistically after a national disaster type event. It could be considered to be original research but there is no hypothesis as this is more of a descriptive study.


-Research question well defined, relevant & meaningful. It is stated how research fills an identified knowledge gap.

L 66– The lead sentence of the paragraph and the concluding one give the general purpose, but the paragraph should be a bit more succinct.

-Rigorous investigation performed to a high technical & ethical standard.
This descriptive study collected that best data available at the time.

-Methods described with sufficient detail & information to replicate.


-A figure with a map of the study area location including its extent and the location of the middens by number would be useful.

-How they measured conch should be explained more detailed than just "standard morphometric measurements". One of the papers by Appeldoorn or other source should be referenced. In particular, the method of measuring LT should be defined as method can easily impact measurements.

-The reader needs a better understanding of the rationale for the conch size classifications other than just matching the earlier study. For clarity, add size category initials (from Clark et al. and raw data file) to the legend on Fig. 4.

-The topic of legal harvest vs. sexual maturity should be brought up in the Methods section. Otherwise, it's not really clear why the SL and LT groups are presented as they are in Fig. 4.

-Other literature to be consulted include: Stoner, AW, Mueller, KW et al. Maturation and Age of Queen Conch: Urgent need for changes in harvest criteria. Fish. Res., 131: 76-84 (2012). Also a technical paper for the Bahamas Ministry of Ag and Marine Resources entitled, “Queen Conch Harvest in The Bahamas during the Last Decade: Shell Middens Provide More Evidence of Overfishing.”

-L 80-81 “Discarded shell middens are assumed to be an accurate representation of fisheries landings.” My interpretation of the referenced paper does not give support for that statement and based on personal observations, there are numerous shells left under the water.

Validity of the findings

2-Experimental Design

-Original primary research within Aims and Scope of the journal.
Subsistence fishing data is rare in the Bahamas so this is unique snapshot of such data collected opportunistically after a national disaster type event. It could be considered to be original research but there is no hypothesis as this is more of a descriptive study.




-Research question well defined, relevant & meaningful. It is stated how research fills an identified knowledge gap.

L 66– The lead sentence of the paragraph and the concluding one give the general purpose, but the paragraph should be a bit more succinct.

-Rigorous investigation performed to a high technical & ethical standard.
This descriptive study collected that best data available at the time.

-Methods described with sufficient detail & information to replicate.


-A figure with a map of the study area location including its extent and the location of the middens by number would be useful.

-How they measured conch should be explained more detailed than just "standard morphometric measurements". One of the papers by Appeldoorn or other source should be referenced. In particular, the method of measuring LT should be defined as method can easily impact measurements.

-The reader needs a better understanding of the rationale for the conch size classifications other than just matching the earlier study. For clarity, add size category initials (from Clark et al. and raw data file) to the legend on Fig. 4.

-The topic of legal harvest vs. sexual maturity should be brought up in the Methods section. Otherwise, it's not really clear why the SL and LT groups are presented as they are in Fig. 4.

-Other literature to be consulted include: Stoner, AW, Mueller, KW et al. Maturation and Age of Queen Conch: Urgent need for changes in harvest criteria. Fish. Res., 131: 76-84 (2012). Also a technical paper for the Bahamas Ministry of Ag and Marine Resources entitled, “Queen Conch Harvest in The Bahamas during the Last Decade: Shell Middens Provide More Evidence of Overfishing.”

-L 80-81 “Discarded shell middens are assumed to be an accurate representation of fisheries landings.” My interpretation of the referenced paper does not give support for that statement and based on personal observations, there are numerous shells left under the water.

Additional comments

This study was undertaken as a rapid review of subsistence fishing of queen conch in The Bahamas in response to COVID-19 restrictions and as such, is unique. It is generally well researched and developed into the context of the existing literature (but see note below on literature that needs to be consulted).
The manuscript borders between being a research paper and a case study and as such could be reduced to a shorter note.

---

## Round 0.2 · Minor Revisions

· Academic Editor

Minor Revisions

Please make sure to address all the reviewers comments. One reviewer suggested to reduce the article to a shorter note. Please consider this as an option to streamline the manuscript and enhance important message.

---

## Round 0.3 · Minor Revisions

· Academic Editor

Minor Revisions

The manuscript presents a unique case of the impact of the COVID-19 pandemic on subsistence fisheries for Queen conch that a broad audience will appreciate. I have made editorial edits on the ms. Please take them in account before submission.

---

## Round 0.4 · accepted · Accept

· Academic Editor

Accept

The original Academic Editor is no longer available and I have taken over handling your submission.

I am sorry for the delay in the decision and am pleased to accept your work. I believe it is a unique and important contribution to the literature.

In the proof stage, please move the reference (Maxwell et al.) on line 53 to a different location, as it currently looks like a species authority, and ensure the word "another" on line 58 is not in italics.